# Neural activity ramps in frontal cortex signal extended motivation during learning

Josue M Regalado, Ariadna Corredera Asensio, Theresa Haunold, Andrew C Toader, Yan Ran Li, Lauren A Neal, Priyamvada Rajasethupathy*

Laboratory of Neural Dynamics & Cognition, The Rockefeller University, New York, United States

**\*For correspondence:**
priya@rockefeller.edu

**Competing interest:** The authors declare that no competing interests exist.

**Abstract** Learning requires the ability to link actions to outcomes. How motivation facilitates learning is not well understood. We designed a behavioral task in which mice self-initiate trials to learn cue-reward contingencies and found that the anterior cingulate region of the prefrontal cortex (ACC) contains motivation-related signals to maximize rewards. In particular, we found that ACC neural activity was consistently tied to trial initiations where mice seek to leave unrewarded cues to reach reward-associated cues. Notably, this neural signal persisted over consecutive unrewarded cues until reward-associated cues were reached, and was required for learning. To determine how ACC inherits this motivational signal we performed projection-specific photometry recordings from several inputs to ACC during learning. In doing so, we identified a ramp in bulk neural activity in orbitofrontal cortex (OFC)-to-ACC projections as mice received unrewarded cues, which continued ramping across consecutive unrewarded cues, and finally peaked upon reaching a reward-associated cue, thus maintaining an extended motivational state. Cellular resolution imaging of OFC confirmed these neural correlates of motivation, and further delineated separate ensembles of neurons that sequentially tiled the ramp. Together, these results identify a mechanism by which OFC maps out task structure to convey an extended motivational state to ACC to facilitate goal-directed learning.

## eLife assessment

This **important** manuscript provides **compelling** experimental evidence of extended motivational signals encoded in the mouse anterior cingulate cortex (ACC) that are implemented by orbitofrontal cortex (OFC)-to-ACC signaling during learning. The experimental methods used were state-of-the-art. These results will be of interest to those interested in cortical function, learning, and/or motivation.

## Introduction

Animals must sustain an extended motivational state to achieve goal-directed learning. Imagine being hungry in the middle of a busy metropolis with no cellphone battery and no way of searching for the nearest restaurant. The feeling of hunger provides motivation to search for restaurant signs, scan menus, and contemplate what type of food to eat. If it is dinnertime and many restaurants are full, this motivational state (hunger) may persist for hours until a restaurant is selected. Thus, an animal's ability to carry out novel actions based on its desired goals is commonly referred to as goal-directed learning. This learning is of a more deliberate, informed nature than habitual learning, as it is sensitive to the current value of outcomes and can lead to a novel sequence of actions for a desired outcome (*Balleine and Dickinson, 1998*; *Tolman, 1948*; *Pezzulo et al., 2014*).

**eLife digest** Achieving goals takes motivation. An individual may have to complete a task many times for a future reward. For example, an animal may have to forage repeatedly to find food, or a person may have to study to get a good grade on a test. How these complex behaviors are encoded in the brain's wiring is not fully understood.

Patients with injuries to the frontal cortex of the brain display a lack of motivation to pursue goals. This discovery suggests the frontal cortex plays a vital role in motivation and goal-directed behavior. Animal studies show that part of their brain's frontal cortex, the anterior cingulate cortex (ACC), helps them stay motivated and put extra effort into achieving goals. Yet, scientists wonder how particular actions are associated with specific goals and suspect the orbital frontal cortex (OFC) contains the blueprint to support this association.

Regalado et al. show that the OFC and ACC work together during goal-seeking behavior in mice. In the experiments, mice learned to complete a task to achieve a sugar water reward. As the mice were learning, Regalado et al. recorded activity in the ACC and found that the ACC is active during goal-seeking behavior. They also discovered that the activity of neurons in the OFC increased the longer mice went without receiving a reward, up until the reward was achieved, signaling a motivational state. Animals not motivated enough to maximize their rewards did not have an increased OFC activity. The experiments also showed that the motivational signals in the OFC were conveyed to ACC to support goal-directed learning, especially linking actions to positive future outcomes.

The experiments help explain how an increase in neuronal activity in the OFC helps to increase motivation and goal-seeking behavior supported by the ACC. More studies will help scientists learn more about these processes and develop drugs or other therapies that can help people who have learning difficulties or struggle with motivation because of an injury or mental illness.

Goal-directed learning often requires the ability to maintain an extended motivational state even in the midst of distracting and competing external variables (*Miller and Cohen, 2001*; *Shenhav et al., 2013*). This function has been long proposed to be carried out by the prefrontal cortex (PFC), as patients with PFC lesions struggle to perform tasks that require maintaining a motivational and goal-directed state, in the midst of competing sensory information, such as the Stroop task or the Wisconsin Card Sorting Task (*Stroop, 1935*; *Yuan and Raz, 2014*; *D'Esposito and Postle, 2015*; *Milner, 1963*; *Pardo et al., 1990*; *Shallice and Burgess, 1991*). In particular, the anterior cingulate cortex (ACC) has been implicated in action selection over long timescales that are influenced by a variety of motivational factors, such as the value and effort required for each outcome (*Shenhav et al., 2013*; *Hauber and Sommer, 2009*; *Hillman and Bilkey, 2010*; *Wallis and Kennerley, 2011*; *Cowen et al., 2012*; *Shenhav et al., 2016*). For instance, when animals are given two choice options: one in which high effort leads to high rewards, and one in which low effort leads to low rewards, animals learn to exploit the high-effort, high-reward option (*Walton et al., 2003*; *Schweimer et al., 2005*). Impairments to the ACC results in animals failing to accurately allocate motivation toward strategies that maximize reward (*Amiez et al., 2006*; *Kennerley et al., 2006*). Single-unit recordings from ACC have shown that neurons encode for choices that require effort with a higher payoff, giving support for the hypothesis that this region is important for action-outcome associations and allocating resources for learning and for the maximization of reward over long timescales (*Hillman and Bilkey, 2010*; *Monosov et al., 2020*; *Holroyd and Yeung, 2012*; *Hillman and Bilkey, 2012*). While the precise functions of ACC are still debated, its role in goal-directed learning is widely accepted (*Shenhav et al., 2013*; *Holroyd and Yeung, 2012*; *Botvinick et al., 2001*; *Heilbronner and Hayden, 2016*; *Rushworth et al., 2012*).

To provide deeper mechanistic insight into how ACC encodes an extended motivational state to facilitate goal-directed learning, we sought to track how animals learn to adjust their behavior over days-long timescales to maximize reward when cue-reward contingencies change. We designed a task in which mice self-initiate trials and learn to associate cues with reward. Through neural activity recordings during behavior, we found that ACC neural activity was consistently tied to trial initiations where mice seek to leave unrewarded cues to reach a rewarded cue. Subsequently, by recording neural activity from inputs to ACC we identified a ramp in bulk activity in orbitofrontal cortex (OFC)-to-ACC projections as mice continuously existed unrewarded cues, peaking when they finally reached

a rewarded cue, thus tracking an extended motivational state. Finally, cellular resolution imaging of OFC-to-ACC neurons identified populations of neurons that sequentially tile the observed bulk neural activity ramp across unrewarded cue presentations. In particular, neurons that preferentially encoded reward cues, before learning, began to code for unrewarded, cues after learning, including the motivation to exit these rooms to reach more reward-associated cues. Taken together, we identified a mechanism by which OFC neural activity ramps map out task structure and conveys an extended motivational state to ACC to enable goal-directed learning.

## Results

### ACC contains neural correlates of motivation during learning

We began by designing a learning task in which mice self-initiate trials and, upon brief cue exposure (an olfactory and auditory cue), learn to stop to collect a water reward (*Figure 1A and B*, *Figure 1— figure supplement 1A*, Materials and methods). We implemented this task in a head-fixed setting to enable hundreds of trials per session, and millisecond precision in tracking stimulus delivery and behavioral responses (*Figure 1A*). We used 'time to initiate trials' as the primary measure of motivation, and 'total reward obtained' as the primary measure of learning. Due to the self-paced nature of the task (*Figure 1B and C*), we found variation between our mice in how quickly they initiated trials and how many rewards they received per minute (*Figure 1C*). As expected, the faster mice can initiate trials, the more rewards they obtained per minute, providing a strong correlation between motivation and learning (*Figure 1D*).

The ACC has been prominently implicated in motivation and voluntary actions for maximizing reward, so we posited that ACC would contain motivation-related neural activity patterns in our task (*Shenhav et al., 2013*; *Monosov, 2017*; *Kolling et al., 2016*; *Khalighinejad et al., 2020*; *Khalighinejad et al., 2022*). To test this hypothesis, we injected AAV1-CaMKII-GCaMP6f into the ACC and implanted fiber-optic cannulas to record bulk neural activity in ACC during behavior (*Figure 1E*). We observed strong neural responses in ACC that were tuned to reward delivery and trial initiations (*Figure 1F and G*). Notably, the ACC neural signal precedes speed onset in both cases, suggesting that ACC is not tracking speed but rather the motivation to initiate trials (*Figure 1F and G*, *Figure 1— figure supplement 1B*). We sought to determine how prolonged inhibition of ACC would impact motivation and whether this was required for learning and reward maximization. We injected AAV9-CaMKII-hM4D(Gi) into ACC and performed chemogenetic inhibition during a session of clozapine N-oxide (CNO) injection (ACC inhibition session) versus a session of saline injection (control session) (*Figure 1H and I*). We found that ACC inhibition caused mice to have a significant increase in time to initiate trials (*Figure 1I*), which also resulted in a decreased number of rewards received per minute (*Figure 1J*). Furthermore, we found a small, but significant, decrease in speed during trial initiation (but not overall session speed), suggesting that ACC inhibition might also impair vigor of movements during trial initiations (*Figure 1—figure supplement 1C*). Thus, we developed a self-paced behavioral task where mice learned cue-reward contingencies, and identified motivation-related signals in ACC that were required for learning to maximize rewards.

### ACC contains neural correlates of extended motivation during learning

We next sought to increase the motivational demand during learning. We thus extended our task by training mice to learn two sets of cue-outcome relationships, where one cue-set (olfactory+auditory) is associated with a sucrose water reward (hereafter referred to as 'R' cues), whereas the other cue-set is associated with no-reward ('N' cues). Since mice have been shaped to stop during cue presentations (*Figure 1*), it is now effortful for them to learn to continue running through the N cues so that they can reach more R cues, and thus maximize their total rewards in a session. Thus, motivation is assessed not only by 'time to initiate trials after R cues', as before, but now also the more effortful measure of 'time to initiate trials after N cues' (*Figure 2A*; see Materials and methods). We measured overall learning through differences in their lick rates and speed during presentations, expecting progressive suppression of licking and increases in speed in the N cues compared to the R cues across days. Interestingly, we found that mice learned to suppress licking in the N cues (*Figure 2A*; red arrows on day 2) much earlier than learning to increase speed in N cues (*Figure 2A*; red arrows on day 4; *Video 1*). Across the cohort, on average, this increase in speed during N cues began as early as day 3, after they had

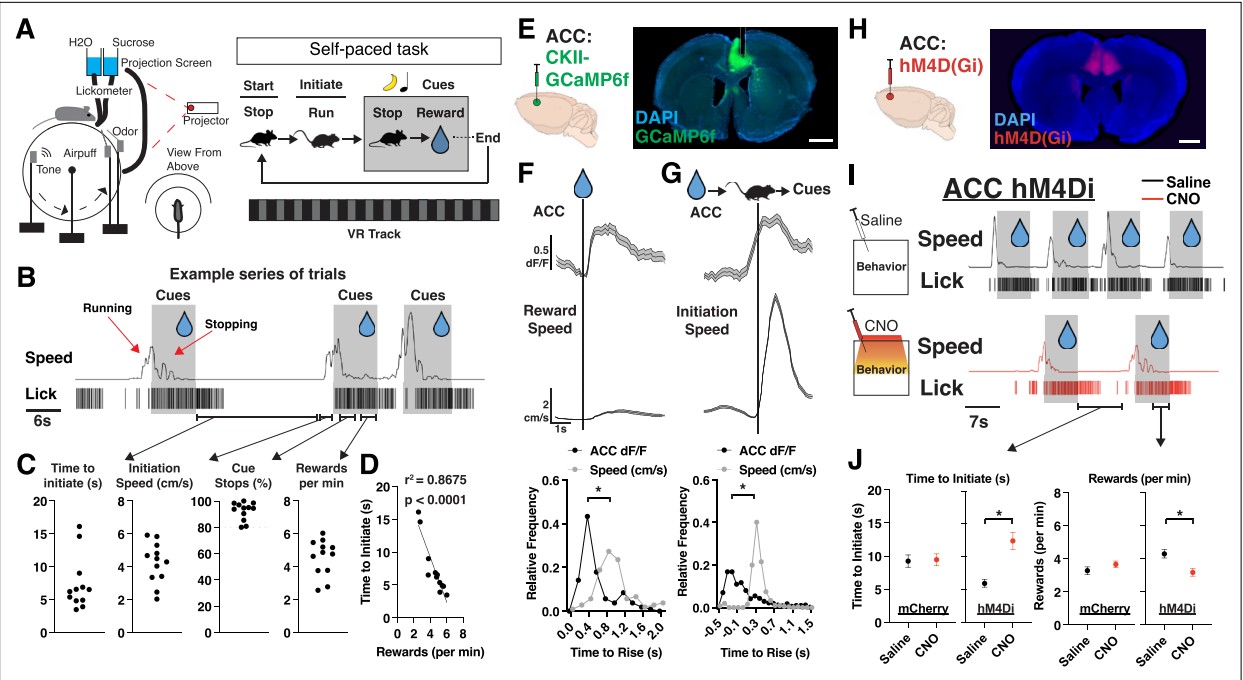

**Figure 1.** Neural activity in anterior cingulate cortex (ACC) signals a motivational state to obtain reward. (**A**) Schematic of virtual reality experimental setup and trial structure. A mouse initiates a trial by running to trigger the onset of cues (olfactory and auditory). After cue onset, a mouse stops to collect a water reward, which ends the trial (see Materials and methods). (**B**) Representative traces of speed and licks from one mouse during a session, with shaded portions corresponding to when cues are on. Red arrows correspond to periods when mice are running to trigger cue onset or stopping to trigger water delivery. Black arrows correspond to sections of a session where we can quantify time to initiate trials, initiation speed, cue stops, and rewards. (**C**) Quantification per mouse of time to initiate a trial (far left; seconds), initiation speed (left; cm/s), % trials in which a stop occurred during cue presentation (right), and rewards received per minute. Individual data points shown (N=12 mice). (**D**) Scatter plots of the mean time (**s**) to initiate a trial plotted alongside rewards received per minute per mouse (N=12 mice). Individual data points shown, with a best fit line, represented by the solid line in the figure. $r^2$=0.8675 and p<0.0001 are determined by linear regression. (**E**) Left: Bulk neural activity recording experimental design. GCaMP6f was injected into the ACC and neural activity was recorded on a fiber photometry setup (see Materials and methods). Right: Brain histology from a representative mouse showing DAPI in blue, GCaMP6f in green, and photometry cannula implantation in ACC (dotted white lines). Scale bar: 1 mm. (**F**) Top: Trial averaged plots of ACC activity (z-scored dF/F) and speed (cm/s) aligned to reward onset. Data are mean (solid line) ± s.e.m. (shaded area). Bottom: Relative frequency plots of the time (**s**) for ACC dF/F or speed to rise above 1 std or 1 cm/s during rewards, respectively (N=105 trials across 12 mice). *p<0.05, paired t-test between time to rise (**s**) between ACC and speed. Data is the frequency of values across time. (**G**) Same as F, but for trial initiations (N=510 trials across 12 mice). (**H**) Injection strategy for DREADDS-based chemogenetic inhibition of ACC during self-paced task. Coronal section from an animal virally injected with AAV1-CamKii-hM4D(Gi) in ACC. DAPI is shown in blue and hM4D(Gi) in red. Scale bar: 1 mm. (**I**) Representative traces of speed and licks from one mouse during the task on a day with saline (top) or clozapine N-oxide (CNO) (bottom) administration 45 min prior to a session, with shaded portions corresponding to when cues are presented. (**J**) Left: Quantification of time (s) to initiate trial (left) across saline and CNO sessions in mCherry-control mice (N=188 trials across 6 mice) and hM4D(Gi)-DREADDs mice (N=215 trials across 4 mice). Right: Same as left but for rewards received per minute in mCherry-control mice (N=60 min across 6 mice) and hM4D(Gi)-DREADDs mice (N=40 min across 4 mice). p=0.8707 for mCherry and *p<0.05 for hM4Di (time to initiate), p=0.2073 for mCherry and *p<0.05 for hM4Di (rewards per min), unpaired t-test between saline and CNO sessions per group. Data are mean ± s.e.m.

The online version of this article includes the following figure supplement(s) for figure 1:

**Figure supplement 1.** Task shaping and speed-related differences between mice and during anterior cingulate cortex (ACC) inhibition.

learned to suppress their licking (day 2), as determined by speed and stop discrimination index (stop DI: % of stops in N – R/all trials) (*Figure 2B and C*, *Figure 2—figure supplement 1A*; see Materials and methods). Finally, there was also a significant correlation between stop DI and rewards obtained per minute, confirming that the development of this behavioral strategy is tied to reward maximization within a given training session (*Figure 2D*).

We next searched for neural correlates of motivation by recording bulk neural activity in ACC as mice performed this task, and aligning neural responses to behavioral frames, focusing on periods when mice learn to run during N cue presentations. As before (as in *Figure 1*), in this two cue-outcome relationship task, we again found that ACC continued to be active during reward delivery and during

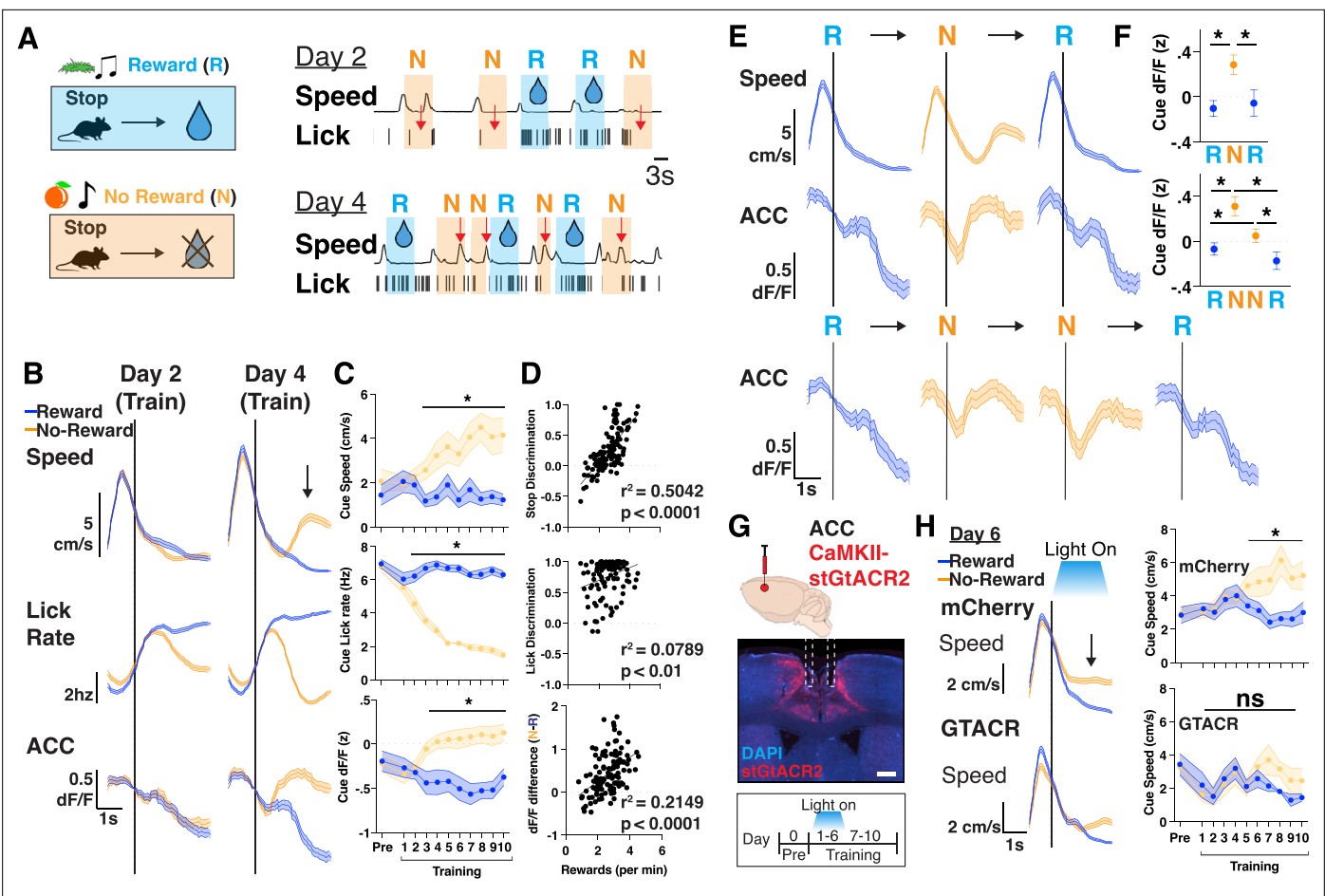

**Figure 2.** Neural activity in anterior cingulate cortex (ACC) scales to match an increased motivational state during learning. (**A**) Top: Schematic of training where mice learn to associate stopping to one set of cues with no water reward ('N') or with water reward ('R'). Bottom: Representative traces of speed and licks from one mouse during a session on training day 2 and day 4, with shaded portions corresponding to when a reward cues (R, blue) or no-reward cues (N, orange) is presented. Red arrow denotes the suppression of licks on day 2, and rise in speed during no-reward cues on day 4. (**B**) Trial averaged speed (cm/s; top), lick rate (Hz; middle), and ACC activity (dF/F z-scored; bottom) aligned to cue presentation across days 2 and 4 of training, separated by reward and no-reward cues (blue vs orange). Black arrow signifies rise in speed after no-reward cue presentation. N=12 mice. Data are mean (dark line) with s.e.m. (shaded area). (**C**) Quantification of average cue speed (cm/s; top), lick rate (Hz; middle), and ACC activity (dF/F z-scored; bottom) across training, separated by reward and no-reward cues (blue vs orange). N=12 mice in each group, data are mean ± s.e.m. *p<0.05, paired t-test between reward and no-reward. (**D**) Scatter plots of rewards per minute vs stop discrimination (top), lick discrimination (middle), or dF/F difference (bottom) for each mouse throughout training (N=120 data points, 12 mice per each of 10 days). Data are individual points with best fit line. $r^2$ and p values are shown, as determined by linear regression. (**E**) Top: Trial averaged speed (cm/s) and ACC activity (dF/F z-scored) aligned to cue presentation across three trials consisting of a reward, no-reward, and reward cue (RNR). Bottom: Trial averaged ACC activity (dF/F z-scored) aligned to cue presentation across four trials consisting of a reward, no-reward, no-reward and reward cue (RNNR). N=12 mice. Data are mean (dark line) with s.e.m. (shaded area). (**F**) Quantification of average cue dF/F activity across RNR and RNNR trial sequences. N=12 mice. *p<0.05, one-way repeated measured ANOVA with post hoc Tukey's multiple comparison test. Data are mean ± s.e.m. (right). (**G**) Top: Injection strategy for AAV1-CaMKII-stGtACR2 into ACC for optogenetic inhibition during training. Middle: Brain histology from a representative mouse showing DAPI in blue, stGtACR2 in red, and photometry cannula implantation in ACC. Scale bar: 1 mm. Bottom: Optogenetic inhibition was targeted to days 1–6 of training and mice were allowed to continue training for days 7–10. (**H**) Left: Trial averaged plots of speed (cm/s) aligned to cue entry on T6 for mCherry controls and GTACR inhibition mice, separated by reward or no-reward cues. Right: Quantification of mean speed during cue presentations. N=8 mice for mCherry, 4 for GTACR early inhibition. *p<0.05, paired t-test.

The online version of this article includes the following figure supplement(s) for figure 2:

**Figure supplement 1.** Lick rate discrimination, anterior cingulate cortex (ACC) learning signal controls, and ACC inhibition lick rate learning.

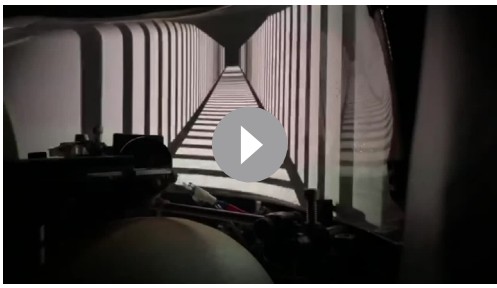

**Video 1.** Behavior during learning. Playback speed: 2×. Shown here is a representative mouse learning to stop in cues that predict reward (blue walls) and run throughout consecutive cue presentations that predict no-reward (yellow walls). Displayed trial sequence order is reward, no-reward, no-reward, and reward (RNNR). https://elifesciences.org/articles/93983/figures#video1

trial initiations (*Figure 2—figure supplement 1B–C*). Additionally, however, in this task we found that ACC began to significantly increase its activity, specifically during N cues, as early as T3, as mice exhibited a learned motivation to leave N cues to reach more R cues (*Figure 2B–D*). As a further confirmation of this result, we investigated ACC's activity during extended motivation across two consecutive N cues and found that ACC activity continued to remain high from the initial N cue presentation until an R cue was reached (*Figure 2E and F*, *Figure 2—figure supplement 1D*). These neural responses, and in particular the dF/F difference in N vs R cues, positively correlated with the amount of reward obtained per minute, linking motivation-related ACC activity to overall learning (*Figure 2D*). Importantly, in all cases, on a trial-by-trial basis, the neural signal preceded the behavioral ramp in speed (*Figure 2E*), and was present even if we restricted our analyses to cue presentations in which mice stopped (*Figure 2—figure supplement 1E*), suggesting a motivational rather than motor response. To further confirm this dissociation, we passively presented both sets of cues to the mice at the end of each training session. As expected, mice did not develop the motivation to run out of N cues (*Figure 2—figure supplement 1F*), and accordingly, the ACC neural activity was no longer different between N and R cues. These results together suggest that ACC encodes a motivation signal to initiate trials, and in particular corresponds to the behavioral measure of running during N cues to reach more R cues, thus facilitating goal-directed learning.

We proceeded to test whether these motivation-related signals in ACC are required for learning. To restrict our inhibition to cue presentation portions of our task, and combat any potential off-target effects of CNO (*Manvich et al., 2018*) from repeated administration across several days of training, we used optogenetic inhibition. We injected AAV1-CaMKII-stGtACR2 bilaterally in ACC to express the inhibitory opsin and delivered light selectively when the mouse received R or N cues, for the first 6 days ('early') or last 4 days ('late') of training (*Figure 2G and H*, *Figure 2—figure supplement 1H*). We found that early ACC inhibition prevented mice from learning to run out of N cues, even though they still learned to suppress their lick rates (*Figure 2H*, *Figure 2—figure supplement 1G*). Late ACC inhibition had no effect on speed or lick rate behavior, as mice continued to run out during N cues while inhibition occurred, suggesting ACC activity does not broadly suppress speed (*Figure 2—figure supplement 1H*). All together, we identified an extended motivation signal in ACC that is required for learning and reward maximization.

## Neural activity in orbitofrontal projections ramps until rewards are reached

The ACC receives projections from disparate regions across the brain that could facilitate the integration of value, internal state, and multisensory information, so we sought to identify how afferent projections may convey motivational signals to ACC during learning (*Fillinger et al., 2017*). We injected rgAAV-hSyn-Cre into ACC and injected AAV1-CAG-FLEX-GCaMP6f in the OFC, anteromedial thalamus (AM), basolateral amygdala (BLA), locus coeruleus (LC), and implanted optical fibers above each region to record neural activity during learning in this task (*Kim et al., 2016*; *Figure 3A*). We first characterized whether the previously observed ACC neural responses during reward delivery and trial initiations were present in any of the inputs to ACC (*Figure 3—figure supplement 1A*). We found that even before learning all projections responded significantly to rewards, and most (OFC$_{ACC}$, AM$_{ACC}$, and LC$_{ACC}$) increased their activity in anticipation of trial initiations (*Figure 3—figure supplement 1A*). Thus, motivation-related signals were broadly present in various projections to ACC.

We then searched for motivation-related neural responses that were specifically tied to learning. To do so, we aligned neural responses to trial initiations after N cues, as mice learned to leave N cues

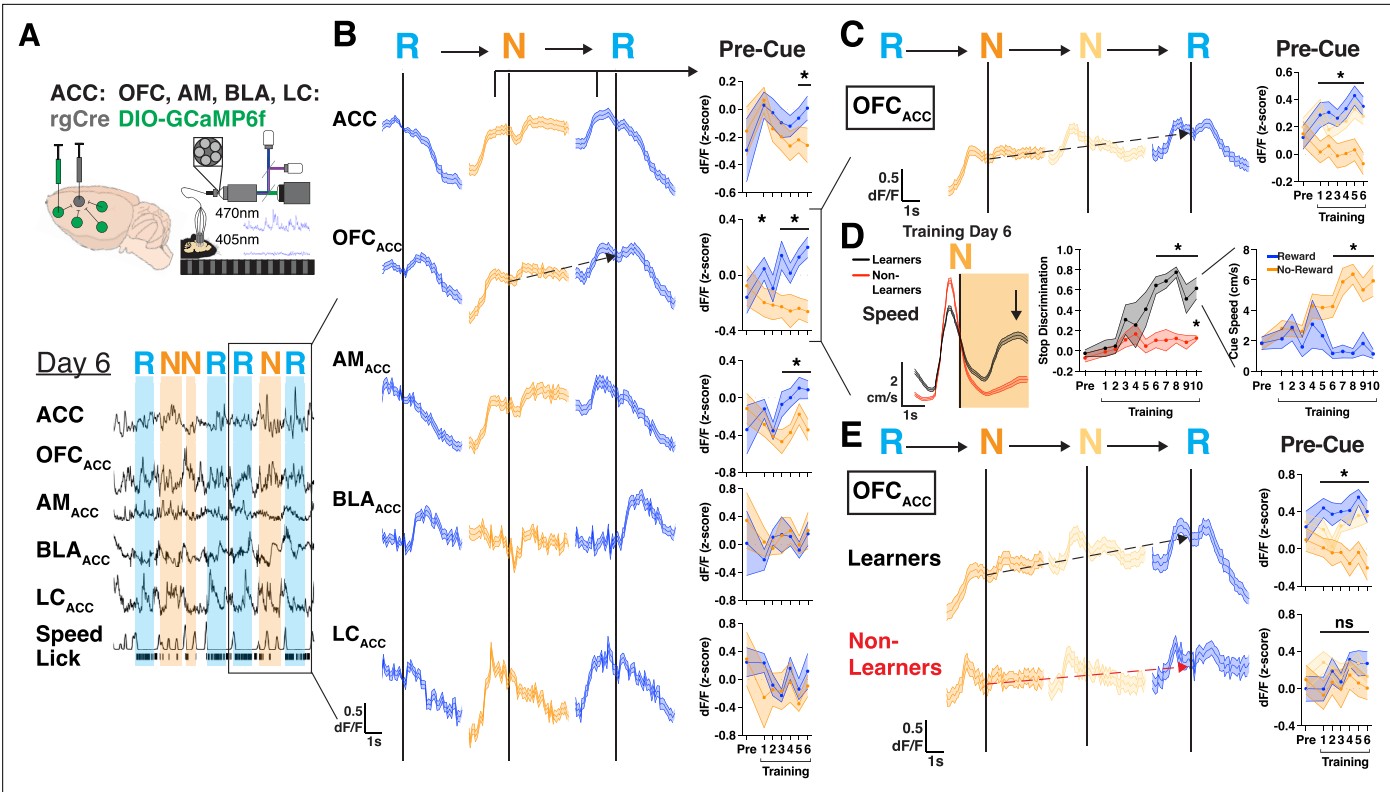

**Figure 3.** Mice with extended motivational states during learning display neural activity ramps in orbitofrontal cortex (OFC). (**A**) Injection strategy and fiber-based photometry setup to record bulk GCaMP6f of projections to anterior cingulate cortex (ACC) from OFC_ACC (orbitofrontal cortex), AM_ACC (anteromedial thalamus), BLA_ACC (basolateral amygdala), or LC_ACC (locus coeruleus). Representative traces for a single mouse showing traces for each region dF/F, speed, and licks. Shaded portions are shown corresponding to when a reward cues (R, blue) or no-reward cues (N, orange) are presented. (**B**) Left: Trial averaged bulk GCaMP6f dF/F of ACC, OFC_ACC, AM_ACC, BLA_ACC, and LC_ACC during a sequence of trials on T6 including reward, no-reward, and reward cues (RNR). Black arrows denote the rise in pre-cue activity from N cue to the following R cue in the RNR sequence. Right: Quantification of pre-cue activity for the N cue and following R cue. Data are mean (solid line) ± s.e.m. (shaded area). N=19, 12, 5, 4 mice, data are mean (solid line) ± s.e.m. (shaded area), *p<0.05, paired t-test between N vs R cues. (**C**) Left: Trial averaged bulk GCaMP6f dF/F of OFC_ACC during a sequence of trials including reward, two no-reward, and reward cues (RNNR). Red arrows denote the rise in pre-cue activity from first N cue to the last R cue in the RNNR sequence. Right: Quantification of pre-cue activity for the first N cue, second N cue, and last R cue. Data are mean (solid line) ± s.e.m. (shaded area). N=19 mice, data are mean (solid line) ± s.e.m. (shaded area), *p<0.05, one-way repeated measures ANOVA with post hoc Tukey's multiple comparison test. (**D**) Left: Speed (cm/s) for 'Learner' (black; reached a DI>0.5 for 3 consecutive days) or 'Non-Learner' (red) mice on training day 6 aligned to no-reward cue onset. Middle: Discrimination index for each group of mice throughout training. Right: Speed during reward and no-reward cues for 'Learner' mice. N=7 ('Learner') and 9 ('Non-Learner') mice. Data are mean (solid line) ± s.e.m. (shaded area), *p<0.05, unpaired t-test between Learner and Non-Learner DI (middle), paired t-test between reward and no-reward cues (right). (**E**) Left: Trial averaged bulk GCaMP6f dF/F of OFC_ACC during a sequence of trials including reward, two no-reward, and reward cues (RNNR). Black arrows denote the rise in pre-cue activity from first N cue to the last R cue in the RNNR sequence. Red arrows denote the absence of this ramp in Non-Learner mice. Right: Quantification of pre-cue activity for the first N cue, second N cue, and last R cue. Data are mean (solid line) ± s.e.m. (shaded area). N=7 ('Learners') and 9 ('Non-Learner') mice, data are mean (solid line) ± s.e.m. (shaded area), *p<0.05, one-way repeated measures ANOVA with post hoc Tukey's multiple comparison test.

The online version of this article includes the following figure supplement(s) for figure 3:

**Figure supplement 1.** Motivation signals in bulk projection activity, and behavior of learners.

to reach more R cues. We found that both OFC_ACC and AM_ACC had higher baseline activity during trial initiations after no-rewards (*Figure 3—figure supplement 1B, C*). To further understand this higher activity after no-rewards we analyzed sequences of 'RNR' trials which contained reward, no-reward, and reward cues (*Figure 3B*). We observed a rise in OFC_ACC activity prior to the N cue presentation that continued to rise until an R cue was reached (black dotted arrow; *Figure 3B*). We quantified this motivational signal as a difference in pre-cue activity between N and R cues in RNR trial sequences across days and found that this difference emerged at the time of learning (~T3) and closely tracked performance of the learned behavior (T3-T6) for OFC_ACC and AM_ACC, but not BLA_ACC or LC_ACC (*Figure 3B*). To further build confidence in these results, we asked whether this continuous rise in OFC_ACC activity

in RNR sequences would be further extended in RNNR sequences. Indeed, OFC$_{ACC}$ activity continued ramping across two consecutive N trials, exhibiting higher pre-cue activity upon entering an R cue after two versus one N (black dotted line, *Figure 3C*).

To more directly determine whether this motivational ramp signal in OFC$_{ACC}$ is tied to learning, we separated our mice into two groups, one that learned the task ('Learners', stop DI>0.5 for at least 3 consecutive days) and one that did not learn ('Non-Learners') (*Figure 3D*, *Figure 3—figure supplement 1D*). The Learners reached a high DI by T6, which persisted throughout the rest of training, whereas the 'Non-Learners' only reached a significantly higher DI by T10 (*Figure 3D*). Both subsets of mice still learned to discriminate with licking at comparable rates (*Figure 3—figure supplement 1E*). When we compared OFC$_{ACC}$ activity in an RNNR sequence of trials, we found that only Learners exhibited a significant ramp in neural activity from the first N cue to the final R cue presentation, which emerged coincidental with behavioral learning and persisted for the remaining days of training (*Figure 3E*). Together, we identify projection activity in OFC that ramps across N cues until an R cue is reached that is specifically tied to the development of a learned goal-directed behavior.

## Orbitofrontal projection neurons tile unrewarded trials until rewards are reached

Given that we identified a ramp in OFC$_{ACC}$ bulk neural activity during NNR sequences (*Figure 3*), we sought to determine whether a single persistently active population or a sequence of tiled neurons underlies this ramp. We thus performed real-time cellular resolution imaging of OFC projections to ACC by injecting rgAAV-hSyn-Cre into ACC and AAV1-CAG-FLEX-GCaMP6f in OFC (*Figure 4A*). We implanted a gradient-index (GRIN) lens above OFC and imaged the region under a two-photon microscope as mice performed the learning task (*Figure 4A*). We focused our analysis on days where behavioral learning emerged (*Figure 4B*), and on NNR trial sequences to find an underlying cellular mechanism to the previously observed photometry results (*Figure 4C*). We found individual neurons that were uniquely active across the first N, second N, or R cue, thereby tiling the sequence of NNR trials (*Figure 4D and E*). We further found that an increasing number of neurons were active along the sequence of NNR trials and most prominently before learning (*Figure 4F*, *Figure 4—figure supplement 1A–C*). Thus, collectively, as an ensemble, these neurons ramp consecutive N cues and peak upon reaching R cues.

To determine how these NNR ensembles facilitate learning we tracked the same population of neurons 'before' and 'after' learning (Stop DI>0.4; *Figure 4G*, *Figure 4—figure supplement 1D*). We identified an ensemble of neurons that were uniquely responsive to R cues preceded by 2 N cues, before learning, and characterized their responses after learning. Interestingly, these neurons were no longer responsive to R cue onset but rather to pre-R cue activity, which then became progressively more responsive to the preceding N cue onset, aligning with the learned behavioral transition of mice leaving N cues to reach R cues (*Figure 4G and H*). To determine whether OFC$_{ACC}$ activity ramps were required for learning, we optogenetically inhibited these projections bilaterally by injecting rgAAV-hSyn-Cre into ACC and AAV1-hSyn-SIO-stGtACR2 into OFC and delivering light only on R or N cues. We then specifically assessed whether previous trial history affected behavioral responses on the current cue condition (*Figure 4I and J*). Interestingly, while both mCherry control and OFC$_{ACC}$ inhibition cohorts could increase their speed during N cues following an R cue, OFC$_{ACC}$ mice were impaired in doing so if the N cue was followed by an N cue (*Figure 4I and J*, *Figure 4—figure supplement 1E*). Taken together, these data demonstrate that ensembles of neurons progressively tile the OFC motivational ramp, and that the initial reward responsive neurons become progressively linked to unrewarded cues after learning, thus effectively linking actions to outcomes to maximize rewards (*Figure 4—figure supplement 1F*).

## Discussion

In this study, we developed a self-paced cue-outcome learning task to determine how mice extend their motivational state to maximize reward over long timescales. We identify the ACC as broadly critical to maximizing reward in our task, especially as mice learn to run out of unrewarded cues. We found that upstream inputs to ACC from OFC sustain a ramp-like increase in activity through consecutive unrewarded cues until mice reach rewarded cues. Cellular resolution imaging of OFC projection

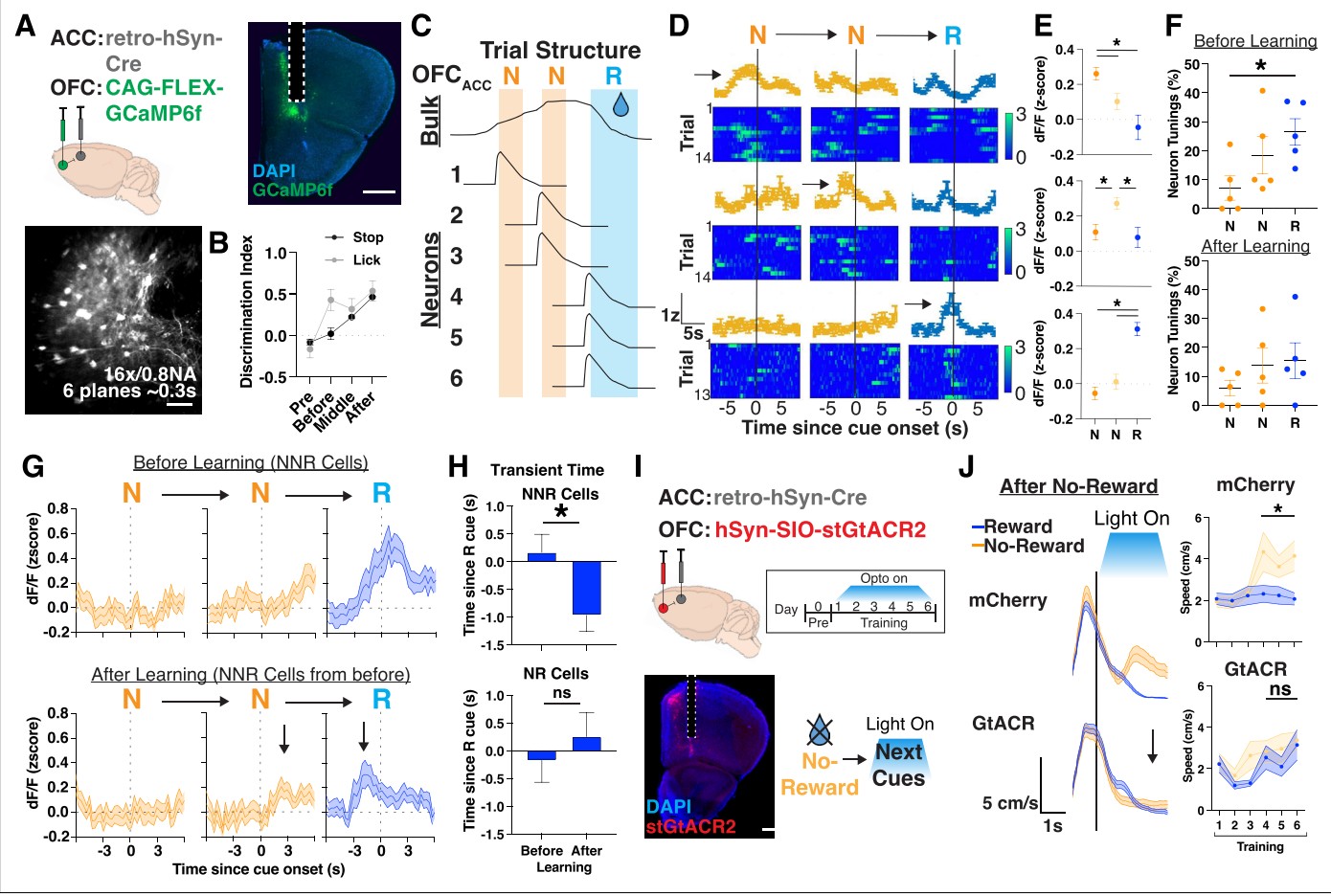

**Figure 4.** Orbitofrontal cortex (OFC) projection neurons tile sequences of trials with no-rewards. (**A**) Injection strategy (top left), histology (top right; scale bar, 1 mm) and z-projection images of two-photon recording (bottom left; mean over time; scale bars, 200 μm) of GCaMP expressing OFC projection neurons with gradient-index (GRIN) implants. Bottom right: Sequence of trials with z-scored dF/F for individual neurons, with shaded portions corresponding to when a reward cues (R, blue) or no-reward cues (N, orange) are presented. Red arrow denotes a dF/F transient occurring after two consecutive N cues. (**B**) Stop (black) or lick (gray; see Materials and methods) discrimination index on the first day stop DI reaches >0.4 ('after') and the two previous days ('before' and 'middle'). N=5 mice. (**C**) Schematic of OFC$_{ACC}$ bulk activity based on *Figure 3* results and potential single neuron findings that tile a sequence of trials with two no-rewards followed by a reward cue presentation (NNR). (**D**) Representative neurons with tunings (std>0.75 for 3 s prior to or after cue presentation) to separate cues in an NNR trial sequence. Trial averaged activity of an N (top), NN (middle), and NNR (bottom) neuron with heat map showing individual trial responses. (**E**) Quantification of neurons tuned to separate cues within an NNR trial sequence and their activity to all other cues. N=17 (**N**), 18 (NN), 32 (NNR) cells out of 115 cells in total. *p<0.05, one-way repeated measures ANOVA with post hoc Tukey's multiple comparison test. (**F**) Percentage of neurons tuned to different cues in an NNR trial sequence before (top) or after (bottom) training. N=5 mice. *p<0.05, one-way repeated measures ANOVA with post hoc Tukey's multiple comparison test. (**G**) Ensemble average plots of neurons tuned to R cues after two consecutive N cue presentations (NNR cells) before learning (top) and their activity after learning (bottom). Black arrows denote the rise in activity prior to R cues after learning. N=18 NNR cells out of 81 cells tracked across days. (**H**) Quantification of transient time (s) since R cue onset for neurons tracked across days. N=132, 170 transient events before and after learning across 18 NNR cells and 105, 59 transient events before and after learning across 12 NR cells. *p<0.05, unpaired t-test. (**I**) Left: Injection strategy for AAV1-hSyn-SIO-stGtACR2 into OFC$_{ACC}$ for optogenetic inhibition during training. Optogenetic inhibition was targeted to training for 6 days. Right: Brain histology from a representative mouse showing DAPI in blue, stGtACR2 in red, and photometry cannula implantation in ACC. Scale bar: 1 mm. (**J**) Left: Mean animal speed (cm/s) aligned to cue zone entry after no-reward on T6 for mCherry control or GtACR mice. Black arrow signifies lack of speed increase during N cues. Right: Quantification of mean change speed in cue zone after no-reward, assessed separately for each cue presentation. N=10 mice for mCherry and 13 mice for GtACR, *p<0.05, paired t-test.

The online version of this article includes the following figure supplement(s) for figure 4:

**Figure supplement 1.** Neuron tunings to NNR task structure and inhibition of OFC$_{ACC}$ neurons.

neurons revealed ensembles of neurons that tile the motivational ramp, and a progressive shift in ensemble tuning during learning such that neurons initially encoding for reward become progressively linked to motivated actions, i.e., trial initiations to reach more rewards. We therefore present a model where OFC contains neurons that increasingly link reward to motivated behaviors, conveying a motivational ramp to ACC, to facilitate learning and reward maximization (*Figure 4C*, *Figure 4—figure supplement 1F*).

The OFC has been implicated in guiding adaptive, flexible behavior by signaling information about future outcomes (*Rudebeck et al., 2013*; *Montague and Berns, 2002*; *Mainen and Kepecs, 2009*; *Rich and Wallis, 2016*; *Padoa-Schioppa and Conen, 2017*). One view sees OFC's function as encoding for the value of the outcomes of events, with various neural correlates having been found for value-guided behavior. Another view sees OFC's function more as building a model of the causal relationships between events, which may or may not entail value assessments, into a cognitive map (*Behrens et al., 2018*). Indeed, OFC neurons have been found to encode sensory-sensory associations even prior to any kind of learning (*Sadacca et al., 2018*). A way to link both perspectives into a single account has been to view value and a cognitive map as occurring along a spectrum, where inferring value onto outcomes hinges upon a map that is created. We have found that mice learn to run out during N cues to more quickly reach R cues, thereby acquiring more rewards over a training session. This behavior can be viewed as both value-guided, as the mouse suppresses their lick rate during N cues, and also requiring a mental model of the environment, as running occurs with the expectation of reaching R cues in the future. Indeed, the pseudorandom trial structure ensures that N cues will be presented no more than two times in a row, such that after two N cues an R cue is guaranteed (see Materials and methods). We thus parsimoniously position OFC as functioning in model-based behaviors, and in the accurate planning of actions based on the learned transition structure of a task (*Drummond and Niv, 2020*).

We linked the ramp-like increase in neural activity in OFC to motivation, but several questions still remain about how motivation is computed and why it would be represented as a ramp. Motivation could be computed as a combination of several variables such as time since last reward, value of reward, and effort to reach future rewards. Future theory-driven analyses could determine how motivation is computed, and whether individual variables of time, value, and effort are encoded as clusters of similar tuned neurons, or mixed and collectively represented at the population level. In either case, it is likely that a combined map of task space and value information carried by OFC are being used to inform downstream regions, such as ACC, for adjusting behavior.

The ACC has been shown to carry information necessary for switching or staying with current behaviors during decision-making and learning in order to maximize rewards and minimize threats or punishments (*Shenhav et al., 2013*; *Monosov, 2017*; *Kolling et al., 2016*). We posit that ACC reads information from OFC about task structure and value to perform computations relevant to allocating behavioral control. We have seen this through our findings that ACC is important for learned behaviors associated with maximizing rewards in our self-paced learning task. We compare the decision to run during N cues to a foraging decision to leave a patch to find alternative options, and ACC's importance in the development of this behavior is reminiscent to signals previously described at the time a foraging decision is reached (*Blanchard and Hayden, 2014*; *Hayden et al., 2011*). We found inhibition of ACC activity affected the development of running during N cues, effectively diminishing an animals' ability to strategy switch (*Kennerley et al., 2006*; *Akam et al., 2021*; *Sarafyazd and Jazayeri, 2019*; *Tervo et al., 2014*). While we did not perform single-cell imaging of ACC in our task, we hypothesize that individual ACC neurons could encode the distribution of actions/opportunities (*Klein-Flügge et al., 2022*) (i.e. stop, run, lick, suppress lick) taken during R or N cues. ACC neurons could compute the relative value of the action taken such that more ACC neurons become recruited once mice learn to run out of N cues. The sustained increase in bulk ACC activity across N cue trials (*Figure 2*) could come from a stable sequence of individual neurons that encode the timescale of the actions taken. In this way, OFC projections would encode current motivation across N cues before learning, which then triggers ACC to compute the value-based actions. Motivational signals in OFC would thus represent state since past rewards/goals, while in ACC these signals represent actions taken to pursue rewards/goals in the future.

Here, we studied learning as a systems-level process guided by top-down signals that maintain a motivational state. Our work showed the recruitment of multiple frontal cortical areas in this process,

which is to be expected as animals are required to build, maintain, and use representations of task structure and value to drive learned, motivated behaviors (*Klein-Flügge et al., 2022*). Future work can build upon the task we developed here to determine how the frontal cortex maintains motivational states across many more cue-outcome associations, and how these associations may dynamically change across time (*Izquierdo et al., 2017*). Lastly, a more synaptic-level approach into how ACC integrates information from upstream regions during learning could reveal important micro-circuit computations, molecular or structural changes during motivational states and learning (*Thornquist et al., 2020*; *Peters et al., 2017*), and potential mechanisms underlying seconds-long behavioral timescale learning rules (*Bittner et al., 2017*).

# Materials and methods

**Key resources table**

| Reagent type (species) or resource | Designation | Source or reference | Identifiers | Additional information |
|---|---|---|---|---|
| Strain, strain background (AAV) | AAV1-CaMKIIa-GCaMP6f | Upenn Vector Core | Addgene#100834 | |
| Strain, strain background (AAV) | AAV1-CAG-FLEX-GCaMP6f | Douglas Kim | Addgene#100835 | |
| Strain, strain background (AAV) | AAV1-CKIIa-stGtACR2-FusionRed | Ofer Yizhar | Addgene#105669 | |
| Strain, strain background (AAV) | AAV9-CaMKIIa-hM4D(Gi)-mCherry | Bryan Roth | Addgene#50477 | |
| Strain, strain background (AAV) | AAV9-CaMKIIa- mCherry | Bryan Roth | Addgene#114469 | |
| Strain, strain background (AAV) | rgAAV-hSyn-Cre | James Wilson | Addgene#105553 | |
| Strain, strain background (AAV) | AAV1-hSyn1-SIO-stGtACR2 | Ofer Yizhar | Addgene#105677 | |
| Strain, strain background (AAV) | AAV9-hSyn-DIO-mCherry | Bryan Roth | Addgene#50459 | |

## Mice

All procedures were done in accordance with guidelines derived from and approved by the Institutional Animal Care and Use Committees (protocol #22,087-H) at The Rockefeller University. Animals used were 8- to 10-week-old naive male C57BL/6J mice (Jackson Laboratory, Strain #000664) at the time of surgery. Mice were group housed (3–5 per cage) with ad libitum food and water, unless mice were water restricted for behavioral assays, in which case they were given 1 mL water per day. Body weight was monitored daily to ensure it was maintained above 80% of the pre-restriction measurement. Surgical procedures and viral injections were carried out in mice under protocols approved by Rockefeller University IACUC and were performed in mice anesthetized with 2% isoflurane using a stereotactic apparatus (Kopf).

## Surgical procedures

Puralube vet ointment was applied to the eyes and 0.2 mg/kg meloxicam was administered intraperitoneally using a 1 mL syringe. Hair from the scalp was trimmed, and the area was sterilized using povidone-iodine swabs and subsequently ethanol swabs. An incision covering the anteroposterior extent was made to allow access to the skull. Injection sites were accessed using a dental drill which made 0.5 mm holes through the skull. All virus was injected using a 35 G beveled needle in a 10 µL NanoFil Sub-Microliter Injection syringe (World Precision Instruments) controlled by an injection pump (Harvard Apparatus) at a rate of 100 nL/min. After all viral delivery, an additional 5–10 min delay was applied to avoid backflush before slowly removing the injection needle. Animals that required cannulas or GRIN lenses were implanted immediately following viral injection. Following surgery, mice were allowed to recover in a single housed cage for up to 12 hr, and were given meloxicam tablets. Mice were typically housed for 3 weeks to allow for adequate expression before behavioral testing or histology.

### Viral injections

- In retrograde tracing experiments, mice were unilaterally injected in ACC (A/P: +1.0, M/L: ±0.35, D/V: −1.4) with rgAAV-CAG-tdT at a volume of 500 nL ($1.0 \times 10^{13}$ vg/mL).

- For fiber photometry experiments, 1 μL of AAV1-CaMKIIa-GCaMP6f (UPenn Viral Core, diluted to 5×10$^{12}$ vg/mL) and rgAAV-hSYN-Cre (1.20×10$^{13}$ vg/mL) was injected into ACC, and AAV1-CAG-FLEX-GcaMP6f (5.0×10$^{12}$ vg/mL) was injected into OFC (A/P: 2.5, M/L: ±1.0, D/V: –2.5), AM (A/P: –0.75, M/L: ±0.5, D/V: –3.55), BLA (A/P: –1.23, M/L: 2.75, D/V: –4.7) and LC (A/P: –5.45, M/L: ±0.85, D/V: –3.7). One week after virus injection, mice were unilaterally implanted with 1.25 mm ferrule-coupled optical fibers (0.48 NA, 400 μm diameter, Doric Lenses) cut to the desired length so that the implantation site is ~0.2 mm dorsal to the injection site.
- For cellular imaging, rgAAV-hSYN-Cre (1.20×10$^{13}$ vg/mL) was injected into ACC and AAV1-CAG-FLEX-GcaMP6f (5.0×10$^{12}$ vg/mL) was injected into OFC.
- For optogenetic inhibition of ACC, AAV1-CaMKIIa-stGtACR2 (1×10$^{13}$ vg/mL) was injected into ACC bilaterally. For controls, AAV1-CaMKIIa-mCherry (7×10$^{12}$ vg/mL) was injected.
- For chemogenetic inhibition of ACC, AAV9-CaMKIIa-hM4D(Gi) (1×10$^{13}$ vg/mL) was injected into ACC bilaterally. For controls, AAV9-CaMKIIa-mCherry (1×10$^{13}$ vg/mL) was injected bilaterally in either region.
- For optogenetic inhibition of OFC-ACC projections, rgAAV-hSYN-Cre (1.20×10$^{13}$ vg/mL) was injected into ACC bilaterally and either AAV1-hSyn1-SIO-stGtACR2 (1.50×10$^{13}$ vg/mL) or AAV9-hSyn-DIO-mCherry (9.0×10$^{12}$ vg/mL) for controls was injected bilaterally into OFC.

## Cannula implants

One week after viral injections, mice undergoing photometry or optogenetic experiments were implanted with fiber-optic cannulas (Doric Lenses). For photometry, mice were unilaterally implanted with 1.25 mm ferrule-coupled optical fibers (0.48 NA, 400 μm diameter, Doric Lenses) cut to the desired length so that the implantation site is ~0.2 mm dorsal to the injection site. For optogenetics, mice were implanted bilaterally with 1.25 mm cannulas (0.22 NA, 200 μm diameter, Doric Lenses). In both cases, cannula implants were slowly lowered using a stereotaxic cannula holder (Doric) at a rate of 1 mm/min until it reached the implantation site, 0.2 mm dorsal to the injection site. In the case of bilateral AM optogenetic inhibition, one cannula was implanted at a 10° angle laterally to the skull in order to prevent stereotactic hindrance. Optic glue (Edmund Optics) was then used to seal the skull/cannula interface and a custom titanium headplate was glued to the skull using adhesive cement (Metabond).

## GRIN lens implants

Immediately following viral injections, mice undergoing calcium imaging were implanted with GRIN lens(es). An incision covering the anteroposterior extent was made, and the skin overlying the skull was cleared. The skull was then cleared and textured using a scalpel. Using a dental drill, 1 mm diameter holes were made at stereotactically determined sites of implantation. Site of drilling was immediately covered using chilled Ringers solution, and using a sterile 28 G × 1.2" insulin syringe and low-pressure vacuum suction, the underlying dura was removed. GRIN lenses (1.0 mm diameter, 4.38 mm length, 0.5 NA from GRINTECH (NEM-1 00-25-1 0-860-5-0.5p)) were wrapped in a 1.25 mm wide custom length stainless steel sleeve (McMaster, catalog #5560K46) using optic glue, made to cover only the part of the lens held external to the brain. With a 0.5 mm burr (Fine Science Tools) attached to a stereotaxic cannula holder, the GRIN was slowly lowered into the brain at a rate of 1 mm/min, ending 0.2 mm dorsal to the injection site. The skull was constantly flushed with chilled 1× PBS. Every time the lens moved 0.8 mm more ventral, it was temporarily retracted 0.4 mm dorsally at the same rate, before continuing down again. We found this especially helpful to maximize the number of observed cells when imaging in deep regions. The skull-sleeve connection was then sealed with glue, and further secured with adhesive cement. A custom titanium headplate was glued to the skull using adhesive cement. Immediately following surgery, mice were injected with 0.2 mg/kg dexamethasone subcutaneously to reduce inflammation.

## Histology

Animals were deeply anesthetized with 5% isoflurane before transcardial perfusion with ice-cold PBS and 4% paraformaldehyde in 0.1 M PB. Brains were then post-fixed by immersion for ~24 hr in the perfusate solution followed by 30% sucrose in 0.1 M PB at 4°C. The fixed tissue was cut into 40 μm coronal sections using a freezing microtome (Leica SM2010R), free-floating sections were stained with DAPI (1:1000 in PBST), and mounted on slides with ProLong Diamond Antifade Mountant (Invitrogen).

Images were taken on a Nikon Inverted Microscope Eclipse Ti-E with a 4×/0.2 NA objective lens. Whole-slide images were stitched with NIS-Elements imaging software and further analyzed in ImageJ and MATLAB.

## Virtual reality system

We used a custom-built virtual reality environment, modified from a previously reported version (*Rajasethupathy et al., 2015*). In brief, a 200-mm-diameter styrofoam ball was axially fixed with a 6-mm-diameter assembly rod (Thorlabs) passing through the center of the ball and resting on 90° post holders (Thorlabs) at each end, allowing free forward and backward rotation of the ball. Mice were head-fixed in place above the center of the ball using a headplate mount. Virtual environments were designed in the virtual reality MATLAB engine ViRMEn (*Aronov and Tank, 2014*). The virtual environment was displayed by back-projection onto white fabric stretched over a clear acrylic hemisphere with a 14-inch diameter placed ~20 cm in front of the center of the mouse. The screen encompasses ~220° of the mouse's field of view and the virtual environment was back-projected onto this screen using a Vamvo Ultra Mini Portable projector. The rotation of the styrofoam ball was recorded by an optical computer mouse (Logitech) that interfaced with ViRMEn to transport the mouse through the virtual reality environment. A National Instruments Data Acquisition (NIDAQ) device was used to send out TTL pulses to trigger the CMOS camera, laser for optogenetics, and the various Arduinos controlling tones, odors, airpuff, lick ports. Additionally, the NIDAQ recorded the capacitance changes of the lick port when licking occurred and the CMOS camera exposures to align lick rate and neural recording/imaging to trial events.

## Behavioral shaping

Starting approximately 3 weeks after surgery, mice were put on a restricted water schedule, receiving 1 mL of water in total per day. Body weight was monitored daily to ensure it was maintained above 80% of the pre-restriction measurement.

After a week of water deprivation, mice were habituated to the styrofoam ball for 2 days by receiving their 1 mL of water per day in head-fixed condition. Then mice were put onto a linear track (vertical gray bars) where water release was contingent on running a short time to trigger the onset of cues (an odor and tone) where they received 5 s of water delivery. Over the course of a session, and in subsequent days, the duration needed to run increased. Once mice could run on the ball for 2 s, we introduced a condition to stop during cue onset to trigger water delivery. Over the course of a session, and in subsequent days, the duration needed to stop increased. If a mouse took longer than 10 min to receive their 1 mL of water on a given day, the duration needed to run and/or stop to get water was repeated on the following day until they could reliably walk on the ball for water under 10 min. Once all mice from a cohort were able to run for 1 s, stop during cues for 3 s, and complete at least 80% of initiated trials, training began.

## Behavioral task

In the final version of the task that was used during all experiments, mice ran down a virtual linear track to trigger contextual cues used to predict the outcome they will receive (~4 µL of sucrose water or no water) if they stop. At the beginning of the linear track, mice self-initiated trials by running (speed >10 cm/s) down a virtual linear track for 1 s. Olfactory and auditory cues would then be presented for 3 s. The auditory cues consisted of 5 kHz or 9 kHz tones outputted by a thin plastic speaker (Adafruit) and olfactory cues consisting of α-pinene or octanol were diluted with mineral oil to 10% and released by a custom-built olfactometer. Both auditory and olfactory cues were outputted by Arduino code under the control of ViRMEn code. The cues for reward were a 5 kHz tone and α-pinene while the cues for no-reward were 9 kHz tone and octanol. Outcome onset would happen under the condition that a mouse dropped their speed below 10 cm/s for at least 1 s before the end of the cues. If the mouse failed to stop for at least 1 s, they would be immediately placed at the start of the linear track and would need to run for 1 s to trigger the next trial start. The outcomes consisted of free access to 10% sucrose water presented by a lickometer (reward) or no water (no-reward), alongside another presentation of contextual cues, for 3 s. Sucrose water output were controlled by Arduino code under the control of ViRMEN code. After the outcome zone mice were transported to the beginning of the linear track to start the next trial. The order of reward and no-reward cue was

pseudorandomly predetermined through code so as to not lead to more than 2 of the same cues presented in a row.

Performance in the task was assessed by average speed and average anticipatory lick rate (during the 3 s of cue presentation) for all reward and no-reward trials in a given session. Prior to training, mice were given a 'preexposure' session where they were exposed to each set of cues, with tap water given upon outcome trigger in both. They were then given 10 days of training (referred to as T1-T10). Each mouse was given 15 min on the ball for each training session, and supplemental water was given to each mouse if they failed to drink 1 mL during a session.

## Behavioral analysis

For behavioral experiments, we quantified several variables within a given session per mouse. We determined how long it took for mice to initiate trials based on how long (s) it took for their speed to be above 1 cm/s for over 1 s after a reward, their speed (cm/s) during trial initiations, the percentage of times they stopped (i.e. their speed (cm/s) was below 1 for at least 1 s by the 3rd second after cue onset) after cue onset within a given session, and how many rewards they received per minute (total rewards per session/minute in a session). For analysis in *Figure 1J* we calculated time to initiate trial on a per trial basis, and rewards per minute on a per minute basis.

We also assessed learning by calculating a stop and normalized lick rate difference, which we refer to as the stop and lick discrimination index (DI). The DI was calculated as follows:

$$\text{Stop DI} = \frac{\text{stops in reward cues - stops in no-reward cues}}{\text{stops in reward cues + stops in no-reward cues}}$$

$$\text{Lick DI} \frac{\text{mean lick rate in reward cues - mean lick rate in aversive cues}}{\text{mean lick rate in reward cues + mean lick rate in aversive cues}}$$

A DI of 1 therefore indicates perfect discrimination, while a DI of 0 indicates chance performance. For all sessions, stops were assessed by whether they triggered the reward or no-reward out-period of the trial (i.e. their speed (cm/s) was below 1 for at least 1 s by the 3rd second after cue onset) and lick rate was calculated in the window of time 3 s after the onset of the cues. Repeated measures ANOVA with Tukey's post hoc test was used to assess learning by comparing to discrimination during preexposure. We separated out cohorts of mice in *Figure 3* based on how well they discriminated with stops. We determined 'Learner' mice by seeing if their stop DI reached above 0.5 for at least 3 consecutive days by Training Day 10, and 'Non-Learner' mice as those mice who did not.

## Chemogenetic inhibition of ACC

For chemogenetic silencing experiments, we injected AAV9-CaMKIIa-hM4D(Gi) (or AAV9-CaMKII-mCherry for controls) bilaterally into ACC. For a week prior to behavioral shaping, mice were habituated to handling and intraperitoneal injections of saline. A solution of CNO was prepared at a concentration of 0.5 mg/mL, and mice were injected at a dosage of 5 mg/kg. Behavioral experiments were conducted 45 min after injection.

## Optogenetic inhibition of ACC

Mice were injected with AAV1-CaMKII-stGtACR2 bilaterally in ACC, while control cohorts were injected with AAV1-CaMKII-mCherry. Cannulas were implanted directly above the injection sites. After 3 weeks, mice underwent shaping as described above, then moved onto training. For inhibition during training, light from a 473 nm laser (15 mW at fiber tip) was delivered through a mono fiber-optic patch cord for 3 s (cue zone followed by reinforcement zone) upon the animal entering the cue zone, throughout the duration of training (~15 min).

## In vivo multi-site photometry recordings
### Photometry setup

A custom multi-fiber photometry setup was built as previously (*Kim et al., 2016*) with some modifications that were incorporated to increase signal to noise, detailed below. Excitation of the 470 nm (imaging) and 405 nm (isosbestic control) wavelengths were provided by LEDs (Thorlabs M470F3, M405FP1) which are collimated into a dichroic mirror holder with a 425 nm long pass filter (Thorlabs DMLP425R). This is coupled to another dichroic mirror holder with a 495 nm long pass dichroic

(Semrock FF495-Di02-25x36) which redirects the excitation light on to a custom branching fiber-optic patchcord of five bundled 400 mm diameter 0.22 NA fibers (BFP(5)_400/430/1100-0.48_3 m_SMA-5xMF1.25, Doric Lenses) using a 10×/0.5 NA Objective lens (Nikon CFI SFluor 10×, Product No. MRF00100). GCaMP6f fluorescence from neurons below the fiber tip in the brain was transmitted via this same cable back to the mini-cube, where it was passed through a GFP emission filter (Semrock FF01-520/35-25), amplified, and focused onto a high sensitivity sCMOS camera (Prime 95b, Photometrics). The multiple branch ends of a branching fiber-optic patch cord were used to collect emission fluorescence from 1.25 mm diameter light weight ferrules (MFC_400/430-0.48_ZF1.25, Doric Lenses) using a mating sleeve (Doric SLEEVE_ZR_1.25). The excitation was alternated between 405 nm and 470 nm by a custom-made JK flip flop which takes the trigger input from the sCMOS and triggers the two excitation LEDs alternatively. Bulk activity signals were collected using Photometrics data acquisition software, Programmable Virtual Camera Access Method (PVCAM).

## Photometry recordings

While mice performed the self-paced contextual learning VR task we recorded bulk calcium signals from five regions: ACC, OFC, AM, BLA, and LC simultaneously. Mice shown in *Figures 1 and 2* with ACC recordings also contained OFC, AM, BLA, and LC recordings, which we compile and show all together in *Figure 3*. We recorded at 18 Hz with excitation wavelengths alternating between 470 nm and 405 nm, capturing calcium dependent and independent signals respectively, resulting in an effective frame rate of 10 Hz.

## Data processing

For analysis, the images captured by the CMOS camera were post-processed using custom MATLAB scripts. Regions of interest were manually drawn for each fiber to extract fluorescence values throughout the experiment. The 405 nm reference trace was scaled to best fit the 470 nm signal using least-squares regression. The normalized change in fluorescence (dF/F) was calculated by subtracting the scaled 405 nm reference trace from the 470 nm signal and dividing that value by the scaled 405 nm reference trace. The true baseline of each dF/F trace was determined and corrected by using the MATLAB function *msbackadj*, estimating the baseline over a 200-frame sliding window, regressing varying baseline values to the window's data points using a spline approximation, then adjusting the baseline in the peak range of the dF/F signal.

## Bulk neural responses

The adjusted calcium signals from photometry were aligned to task events (e.g. cue onset, reward, trial initiation, etc.) in ViRMEn by time-stamping behavioral frames captured through the NIDAQ. Photometry signals from all animals from a given region were z-scored across the entire session. The mean regional responses to task variables (*Figures 1–3*) is the mean of these aligned z-scored signals across all animals, with s.e.m. calculated across all recorded trials. We then sought to quantify the difference in mean average activity patterns observed in response to each cue presentation. To calculate the differential response to reward and trial initiation portions of the task (*Figure 1*), we calculated the time it took for dF/F activity to rise above 1 std and speed to rise above 1 cm/s (*Figure 1F*) or dF/F activity to rise above 0.5 std and speed to rise above 2 cm/s (*Figure 1G*). We zeroed the dF/F activity to the start of reward or 2 s prior to trial initiation.

We then sought to quantify the difference in temporal divergence activity patterns observed in reward or no-reward cue presentation. To calculate the differential response to cue onset we calculated the mean z-scored signal from 0 s to 3 s after cue onset (*Figure 2*). We also quantified the difference in pre-cue activity along sequences of trials with one or two no-reward cues to identify ramps in neural activity between reward cues (*Figure 3*). We zeroed the pre-cue activity of all trials within a given sequence to the activity at the time of the first reward cue and calculated the mean z-scored signal between 2 s before to 0 s before cue onset.

## In vivo cellular resolution calcium imaging

### Imaging setup

For imaging in *Figure 4*, mice were imaged throughout training in 15 min sessions per day. Volumetric imaging was performed using a resonant galvanometer two-photon imaging system (Bruker), with a laser (Insight DS+, Spectra Physics) tuned to 920 nm to excite the calcium indicator, GCaMP6f, through a 16×/0.8 water immersion objective (Nikon) interfacing with an GRIN lens through a few drops of distilled water. Prior to each session, mice were head-fixed and each GRIN lens was carefully cleaned with 70% ethanol. Fluorescence was detected through GaAs photomultiplier tubes using the Prairie View 5.4 acquisition software. Black dental cement was used to build a well around the implant to minimize light entry into the objective from the projector. High-speed z-stacks were collected in the green channel (using a 520/44 bandpass filter, Semrock) at 512 × 512 pixels covering each x–y plane of 800 × 800 mm over a depth of ~150 μm (30 μm apart) by coupling the 30 Hz rapid resonant scanning (x–y) to a z-piezo to achieve ~3.1 Hz per volume. Average beam power measured at the objective during imaging sessions was between 20 mW and 40 mW. An incoming TTL pulse from ViRMEn at the start of behavior enabled time-locking of behavioral epochs to imaging frames with millisecond precision.

### Source extraction

Calcium imaging data for *Figure 4* was acquired by Prairie View 5.4 acquisition software and subsequently processed using the Suite2p toolbox (*Pachitariu et al., 2017*). Motion correction, ROI detection, and neuropil correction were performed as described. Cell identification was verified by manually validating every extracted source. Cell registration across sessions for *Figure 4* was performed with a combination of custom scripts and existing packages (Cell Reg, *Sheintuch et al., 2017*).

### Calculation of single-cell dF/F and transient identification

For each cell detected via automated source extraction, a normalized ΔF/F was calculated and individual $Ca^{2+}$ transients were identified as previously described (*Rajasethupathy et al., 2015*). Briefly, ΔF/F was defined as: $(F − F_{baseline})/F_{baseline}$, where F is the raw output ('F') from the suite2p algorithm, and where $F_{baseline}$ is the baseline fluorescence, calculated as the mean of the fluorescence values for a given cell, continuously acquired over a 20 s sliding window to account for slow timescale changes observed in the fluorescence across the recording session. For all analysis, this dF/F was then normalized by z-scoring the entire time series across a session. To identify statistically significant transients, we first calculated an estimate of the noise for each cell using a custom MATLAB script, with a previously described method (*Toader et al., 2023*; *Yadav et al., 2022*). In essence, we identified the limiting noise cutoff level for a given cell using time periods that are unlikely to contain neural events, and then defined a transient as significant if it reached above at least 3σ of this estimated noise level. A custom MATLAB script using the function 'findpeaks' was used to identify any remaining obvious transients not identified by this method (typically when multiple transients occurred in rapid succession). Additional specifications required transients to persist above this noise level for at least 300 ms (roughly twice the duration of the half-life decay time of GCaMP6f). The transient duration was defined as the first and last frames where the dF/F reached 3σ. The value of dF/F was set to zero outside the duration of every identified transient to minimize effects of residual background fluorescence.

### Single-cell cue tuning

To calculate the tuning of an individual cell in anticipation to or during reward or no-reward cues, we z-scored the trial averaged the activity on a given neuron across all the cue presentations for a given reward or no-reward trial. A cell was considered tuned if the magnitude of its trial averaged z-scored activity was at least 0.75 between 3 s before or after cue onset. To find tuning for cues based on previous trial history, we preselected cues that were preceded by specific combinations of trials.

### Transient time analysis

To calculate the transient times of an individual cell tracked before and after learning, we first preselected cells that are tuned to a particular trial sequence (such as NNR). We then identified the frame when the dF/F value first rises above the noise threshold (see Calculation of single-cell dF/F and

transient identification) 7 s before or after cue onset (such as for the R cue in an NNR tuned cell). We took all the transients for any given cell across all the trials in a session, in case a single cell fired more than one transient.

## Statistical analysis

Sample sizes were selected based on expected variance and effect sizes from the existing literature, and no statistical methods were used to determine sample size a priori. Prior to experiments being performed, mice were randomly assigned to experimental or control groups. The investigator was blinded to all behavioral studies. Data analyses for calcium imaging (in vitro and in vivo datasets) were automated using MATLAB scripts. Statistical tests were performed in MATLAB 2017a, 2021b, or GraphPad Prism.

## Inclusion and diversity

One or more of the authors of this paper self-identifies as an underrepresented ethnic minority in science. One or more of the authors of this paper self-identifies as a member of the LGBTQ+ community. One or more of the authors of this paper received support from a program designed to increase minority representation in science.

## Acknowledgements

We thank Nakul Yadav for discussions surrounding two-photon imaging and resulting analysis. We are grateful to Dr. Vanessa Ruta, Dr. Gaby Maimon, Dr. Zachary Mainen, and Rajasethupathy lab members for helpful discussions related to various aspects of the study. We thank Alessandra Bonito Oliva for help with project coordination and discussions/comments on the manuscript. This work was supported by an NSF Graduate Research Fellowship and HHMI Gilliam fellowship awarded to JMR, and grants from the Searle and Klingenstein foundations and from the National Institutes of Health under award number DP2AG058487 to PR.

## Additional information

### Funding

| Funder | Grant reference number | Author |
| --- | --- | --- |
| National Science Foundation | Graduate Research Fellowship | Josue M Regalado |
| Howard Hughes Medical Institute | Gilliam Fellowship | Josue M Regalado |
| National Institutes of Health | DP2AG058487 | Priyamvada Rajasethupathy |
| Searle Scholars Program | | Priyamvada Rajasethupathy |

The funders had no role in study design, data collection and interpretation, or the decision to submit the work for publication.

### Author contributions

Josue M Regalado, Conceptualization, Resources, Data curation, Software, Formal analysis, Supervision, Funding acquisition, Validation, Investigation, Visualization, Methodology, Writing - original draft, Writing - review and editing; Ariadna Corredera Asensio, Conceptualization, Investigation, Methodology; Theresa Haunold, Investigation, Methodology; Andrew C Toader, Data curation, Formal analysis, Methodology; Yan Ran Li, Lauren A Neal, Investigation; Priyamvada Rajasethupathy, Supervision, Funding acquisition, Project administration, Writing - review and editing

### Author ORCIDs

Josue M Regalado http://orcid.org/0000-0002-6798-3860
Ariadna Corredera Asensio http://orcid.org/0000-0003-4289-4500
Theresa Haunold http://orcid.org/0009-0007-8343-4945

Lauren A Neal  http://orcid.org/0000-0002-0092-2852
Priyamvada Rajasethupathy  http://orcid.org/0000-0001-8741-2347

## Ethics

All procedures were done in accordance with guidelines derived from and approved by the Institutional Animal Care and Use Committees (protocol #22087-H) at The Rockefeller University. Animals used were 8-10 weeks-old naive male C57BL/6J mice (Jackson Laboratory, Strain #000664) at the time of surgery. Mice were group housed (3-5 per cage) with ad libitum food and water, unless mice were water restricted for behavioral assays, in which case they were given 1 mL water per day. Body weight was monitored daily to ensure it was maintained above 80% of the pre-restriction measurement. Surgical procedures and viral injections were carried out in mice under protocols approved by Rockefeller University IACUC and were performed in mice anesthetized with 2% isoflurane using a stereotactic apparatus (Kopf).

Reviewer #1 (Public review): https://doi.org/10.7554/eLife.93983.3.sa1
Reviewer #2 (Public review): https://doi.org/10.7554/eLife.93983.3.sa2
Author response https://doi.org/10.7554/eLife.93983.3.sa3

## Additional files

### Supplementary files
• MDAR checklist

### Data availability

All primary behavioral data and relevant code for data analysis are available at Zenodo (https://zenodo.org/records/11451977) and Github (https://github.com/RajasethupathyLabGitHub/RegaladoElifePaperCode, copy archived at *RajasethupathyLabGitHub, 2024*) without restriction. Source data files contain the summarized data for all plots in Figures 1-4.

The following dataset was generated:

| Author(s) | Year | Dataset title | Dataset URL | Database and Identifier |
| --- | --- | --- | --- | --- |
| Josue R, Ariadna CA, Theresa H, Andrew T, Yan Ran L, Lauren N, Priyamvada R | 2024 | Neural activity ramps in frontal cortex signal extended motivation during learning | https://doi.org/10.5281/zenodo.11451977 | Zenodo, 10.5281/zenodo.11451977 |

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
