## [Editor Report · eLife assessment]

This **important** manuscript provides **compelling** experimental evidence of extended motivational signals encoded in the mouse anterior cingulate cortex (ACC) that are implemented by orbitofrontal cortex (OFC)-to-ACC signaling during learning. The experimental methods used were state-of-the-art. These results will be of interest to those interested in cortical function, learning, and/or motivation.

---

## [Referee Report · Reviewer #1 (Public review)]

This is an interesting report examining activity patterns in mouse ACC and in the OFC neurons projecting to ACC. In addition, the effects of inactivation are examined. In aggregate, the results provide new and interesting information about these two brain areas and they translate motivation into action - a function that it seems intuitively plausible that ACC might perform but, despite this intuition, there have been comparatively few direct tests of the idea and little is known of the specific mechanisms. The study is performed carefully and is written up clearly.

The combination of recording and inactivation/inhibition experiments and the combination of investigation of ACC neurons and of OFC regions projecting to ACC are very impressive.

---

## [Referee Report · Reviewer #2 (Public review)]

Summary:

Regalado et al. studied how an extended motivational state, necessary for maintaining behavioural drive despite unrewarding experiences, could be encoded in the ACC and its potential causal implications for learning discriminatory behaviour and avoiding unrewarding stimuli. They designed a self-initiated learning task and identified bulk neural responses tuned specifically to reward delivery as well as trial initiation. Interestingly, in both cases, neural activity precedes behavioural onset, indicating the encoding of a motivational signal. To investigate the neural encoding of motivational signals during unrewarded, distracting stimuli presentation, they created a discrimination task by introducing 'no reward' cues, during which animals need to learn not to reduce running speed and not engage in licking. Interestingly, with mice learning to increase running speed and reduce licking rates after 'no reward' cues, the preceding ACC activity also gradually increased. Importantly, only the increase in running speed after 'no reward' cues was impaired upon optogenetic inhibition of ACC activity during early training, linking the extended motivational signal in ACC and learning to maximise rewards by actively avoiding distracting and unrewarded stimuli. Such motivational signals could also be observed in OFC-ACC projecting neurons. Especially the continuous ramping of activity upon repeated 'non-reward' cues, which could be exclusively observed in the 'fast learner' subgroup, provides an interesting concept of how an extended motivational signal necessary for learning avoidance of unrewarded stimuli could be implemented in ACC. The shift in the temporal activity of initially reward-responsive neurons towards the preceding 'no reward' cue, provides a potential mechanism linking extended motivation to reward maximisation. This mechanism seems to be particularly important in periods of persistent 'non-reward' cues, as demonstrated in the impairment of running speed increase after two consecutive 'non-reward' cues.

Appraisal:

The authors provide convincing experimental evidence to support their claims of an extended motivational signal encoded in the ACC that is implemented by OFC-ACC signalling and critically involved in learning avoidance of unrewarded stimuli. The newly designed task seems appropriate to identify correlates of relevant cognitive and behavioural variables (e.g. sustained motivation). The combination of recording Ca2+ transients (bulk as well as longitudinal single neuron recordings) to identify potential neural responses and subsequent evaluation of their causal role in establishing and maintaining this persistent motivational state using opto- and pharmacogenetic manipulations is generally accepted.

Impact:

The findings will be valuable for further research on the impact of motivational states on behaviour and cognition. The authors provided a promising concept of how persistent motivational states could be maintained, as well as established a novel, reproducible task assay. While experimental methods used are currently state-of-the-art, theoretical analysis seems to be incomplete/not extensive.

---

## [Author Response]

The following is the authors’ response to the original reviews.

**eLife assessment**
The authors provide convincing experimental evidence of extended motivational signals encoded in the mouse anterior cingulate cortex (ACC) that are implemented by the orbitofrontal cortex (OFC)-to-ACC signaling during learning. The results are valuable to the field of motivation and cognition. The experimental methods used were state-of-the-art. The manuscript would further benefit from theory-driven analyses to inform a mechanistic understanding, particularly for the single-cell calcium imaging results. These results will be of interest to those interested in cortical function, learning, and/or motivation.

We thank the reviewers for their thoughtful reading of our paper and providing constructive feedback. We have made the relevant changes to the manuscript to improve the writing and figures. We provide responses below to each of the reviewer’s comments.

**Reviewer #1 (Public Review):**
(1) An important conclusion (Figure 4) is that when mice are trained to run through no reward (N) cues in order to reach reward (R) cues, the OFC neurons projecting to ACC each respond to different specific events in a manner that ensures that collectively they tile the extended behavioural sequence. What I was less sure of was whether the ACC neurons do the same or not. Figure 3 suggests that on average ACC neurons maintain activity across N cues in order to get to R cues but I was not sure whether this was because all individual neurons did this or whether some had activity patterns like the OFC neurons projecting to ACC.

We agree that it remains uncertain what individual ACC neurons do during the extended behavioral sequence. We now include a few sentences in the discussion about what we hypothesize, as we did not perform the cellular resolution imaging to determine this:

“While we did not perform single-cell imaging of ACC in our task, we hypothesize that individual ACC neurons could encode the distribution of actions/opportunities47 (i.e. stop, run, lick, suppress lick) taken during R or N cues. ACC neurons could compute the relative value of the action taken such that more ACC neurons become recruited once mice learn to run out of N cues. The sustained increase in bulk ACC activity across N cue trials (Figure 2) could come from a stable sequence of individual neurons that encode the timescale of the actions taken. In this way, OFC projections would encode current motivation across N cues before learning, which then triggers ACC to compute the valuebased actions. Motivational signals in OFC would thus represent state since past rewards/goals, while in ACC these signals represent actions taken to pursue rewards/goals in the future.”

(2) Figure 1 versus Figure 2: There does not seem to be a particular motivation for whether chemogenetic inactivation or optogenetic inhibition were used in different experiments. I think that this is not problematic but, if I am wrong and there were specific reasons for performing each experiment in a certain way, then further clarification as to why these decisions were made would be useful. If there is no particular reason, then simply explaining that this is the case might stop readers from seeking explanations.

Thank you for this comment and we agree that clarification on this is important. We performed chemogenetic inhibition of ACC in Figure 1 to take a broad survey of behavioral effects throughout a 40-min long behavioral session, and performed optogenetic inhibition in Figure 2 because we wanted to restrict our inhibition to the few seconds of cue presentation during a behavioral session and across days. Furthermore, we wanted to combat any potential off-target effects that would come from repeated administration of CNO over the several days of training (Manvich et al 2018). We have included a couple sentences on page 4 to clarify this:

“We proceeded to test whether these motivation related signals in ACC are required for learning. To restrict our inhibition to cue presentation portions of our task, and combat any potential off-target effects of CNO31 from repeated administration across several days of training, we used optogenetic inhibition.”

(3) P5, paragraph 2. The authors argue that OFC and anteriomedial (AM) thalamic inputs into ACC are especially important for mediating motivation through N cues in order to reach R cues. Is this based on a statistical comparison between the activity in OFC or AM inputs as opposed to the other inputs?

We determined that OFC and AM thalamic inputs to ACC are particularly important by comparing the pre-cue activity in a reward-no reward-reward trial sequence (RNR; Figure 3B). Specifically, we performed paired t-tests comparing pre-cue activity between N and R cues, and found a statistically significant increase for R cues but only for the OFC and AM inputs, not for the BLA or LC inputs.

(4) P3, paragraph 2. Some papers by Khalighinejad and colleagues (eg Neuron 2020, Current Biology, 2022) might be helpful here in as much as they assess ACC roles in determining action frequency, initiation, and speed and mediating the relationship between reward availability and action frequency and speed.

We thank the reviewer for bringing these relevant papers to our attention. We have included these papers in our citations in this paragraph.

(5) Paragraph 1 "This learning is of a more deliberate, informed nature than habitual learning, as they are sensitive to the current value of outcomes and can lead to a novel sequence of actions for a desired outcome1-3." Should "they" be "it"?

This is correct, we have edited this in the manuscript.

**Reviewer #2 (Public Review):**
Impact:The findings will be valuable for further research on the impact of motivational states on behaviour and cognition. The authors provided a promising concept of how persistent motivational states could be maintained, as well as established a novel, reproducible task assay. While experimental methods used are currently state-of-the-art, theoretical analysis seems to be incomplete/not extensive.We thank the reviewer for these comments. In our paper, we performed single-cell calcium imaging of OFC projection neurons to ACC to build a mechanistic understanding for the bulk ramp-like response we identified in these neurons with photometry. We identified ensembles of neurons that tile sequences of trials that match the bulk response, in particular a subset of neurons that are active at the time a reward (R) cue is reached after 2 no-reward (N) cues. We included a paragraph in the discussion to address future theory-driven analyses to address how computation is achieved by OFC projection neurons:

“We linked the ramp-like increase in neural activity in OFC to motivation, but several questions still remain about how motivation is computed and why it would be represented as a ramp. Motivation could be computed as a combination of several variables such as time since last reward, value of reward, and effort to reach future rewards. Future theorydriven analyses could determine how motivation is computed, and whether individual variables of time, value, and effort, are encoded as clusters of similar tuned neurons, or mixed and collectively represented at the population level. In either case, it is likely that a combined map of task space and value-information carried by OFC are being used to inform downstream regions, such as ACC, for adjusting behavior. ”

**Reviewer #2 (Recommendations for the Authors):**
Overall, the layout of the figures seems a little bit chaotic and makes it hard to understand the boundaries between panels.

We agree that the figure layout could be improved upon to aid the reader in moving from panel to panel. We have edited two of the main figures with layouts that are most irregular (Figures 2 and 4) to help with this.

Figures/text should include the promoters used for protein expression so that readers understand which cell types would be affected.

We have made sure to edit the figures to include the promoter of the viruses we used, and edited the text to include both the AAV serotype and promoter.

Discuss why it is necessary for multiple prefrontal areas to be involved in maintaining motivational signals.

We thank the reviewer for this comment. We believe that prefrontal areas would be recruited as tasks to study motivational states become more complex and require animals to keep track of task structure and perform value-guided actions. We have included a couple sentences in the final paragraph of the discussion about this:

“Our work showed the recruitment of multiple frontal cortical areas in this process, which is to be expected as animals are required to build, maintain, and use representations of task structure and value to drive learned, motivated behaviors47. Future work can build upon the task we developed here to determine how the frontal cortex maintains motivational states across many more cue-outcome associations, and how these associations may dynamically change across time48”.

Additionally, we included a short discussion on how in motivational signals differ between OFC and ACC in our work. We suggest OFC encodes current motivation before and after learning, which then leads ACC to represent learned actions taken and thus have a longer timescale motivational response (see response to Reviewer 1).

Minor: Page 4, Line 1: "increase" instead of "increases".

This is correct, we have edited this in the manuscript.